# Experiences and actions related to living with type 1 diabetes during the COVID-19 pandemic in Norway: a qualitative study conducted during July to December 2020

Karin Pleym,[1,2] Marjolein Memelink Iversen  ,[1] Anders Broström[3,4]

[1]Department of Health and Caring Sciences, Western Norway University of Applied Sciences, Bergen, Vestlandet, Norway
[2]Department of Medicine, Vestre Viken Hospital Trust, Drammen, Norway
[3]Department of Nursing, School of Health and Welfare, Jönköping University, Jönkoping, Sweden
[4]Department of Clinical Neurophysiology, Linköping University Hospital, Linköping, Sweden

**Correspondence to**
Dr Marjolein Memelink Iversen;
marjolein.memelink.iversen@hvl.no

## ABSTRACT

**Objectives** The aim of this study was to describe the experiences of adults with type 1 diabetes (T1DM) during the COVID-19 pandemic in Norway, and what actions they took to cope with the situation.

**Design** An inductive, descriptive design applying the critical incident technique was used to collect qualitative data between July 2020 and December 2020.

**Setting** A strategic selection was made from diabetes specialist outpatient clinics at three different hospitals in eastern Norway. The hospitals, two community hospitals and one university hospital, were situated in both rural and urban areas.

**Participants** Inclusion criteria were people with T1DM aged 18–65 years. Exclusion criteria were pregnancy, people with chronic pulmonary disorders, people with active cancer diseases and people diagnosed with a myocardial infarction or stroke during the previous 6 months. Semistructured individual interviews with 19 people with T1DM were conducted.

**Results** Experiences were categorised into two main areas: 'increased psychosocial burden of T1DM during the COVID-19 pandemic' and 'changed conditions for T1DM treatment during the COVID-19 pandemic'. Uncertainty distress and social consequences from infection control measures contributed to the burden of T1DM. Disrupted T1DM follow-up and altered daily routines created challenges. However, having increased time to focus on T1DM self-management during lockdown represented an improvement. Actions to handle the situation were categorised into two main areas: 'actions to handle psychosocial strain related to T1DM and COVID-19' and 'actions to handle changed conditions for T1DM treatment during the COVID-19 pandemic'.

**Conclusions** Patients experienced an increased psychosocial burden of T1DM and difficulties from a disrupted daily life affecting T1DM self-management routines. Uncertainty-reducing behaviours and actions to adapt to the situation provided a general sense of coping despite these difficulties. Tailored information and follow-up by telephone or video call was emphasised to reduce uncertainly distress and support adequate diabetes T1DM self-management.

### Strengths and limitations of this study

► Three researchers with good knowledge of both diabetes and critical incident technique (CIT) have been involved throughout the study.
► Using the CIT, we were able to focus on both experiences and actions used by people with type 1 diabetes during the COVID-19 pandemic in Norway.
► The majority of the interviews were performed by video call, giving the participants the comfort of familiar environments and contributing to a safe interview situation.
► A possible paradox in the study may be that subjects who suffer from lockdown, usually could be the ones who want to participate in the study.
► Norway had an early complete lockdown prior to the data collection, and relatively few deaths from COVID-19 compared with other European countries.

## INTRODUCTION

The outbreak of SARS-CoV-2-induced COVID-19 was declared a global pandemic by the WHO on 11 March 2020.[1] The COVID-19 disease ranges in severity from asymptomatic or mild cases to severe illness and death.[2 3] Diabetes has been identified as one of several underlying health conditions associated with increased severity and higher mortality rates from COVID-19 illness.[4–9]

In Norway, the government implemented strict interventions to limit the spread of the virus.[10] The infection control measures had significant impacts on the population's everyday life. Previous reports from preceding coronavirus epidemics such as SARS-CoV (SARS) in 2002–2003 and the Middle East respiratory syndrome coronavirus in 2012–2015 showed evidence of increased psychosocial and mental health challenges.[11–13]

Similarly, psychosocial challenges caused by the COVID-19 pandemic have been seen in

the general population.[14–16] The massive media coverage and the general perception of the severity of the situation had potential to cause stress and anxiety. People with type 1 diabetes (T1DM) can be more vulnerable to stress in terms of a direct effect of stress hormones on blood glucose levels and an indirect effect on the person's behaviour related to diabetes self-management.[17] Previous studies have shown that people with T1DM experienced disrupted follow-up from specialist outpatient clinics during the COVID-19 pandemic and challenges affecting T1DM self-management routines.[18–22] Other studies, however, showed improved glycaemic control in periods of lockdown of society.[23–25] A cross-sectional survey from Denmark, in which 2430 adults with diabetes completed an online questionnaire, indicated that people with diabetes have COVID-19-specific worries about their diabetes associated with poorer psychosocial health.[26] In line with these results, diabetes nurses across Europe have reported a significant increase in physical and psychological problems in patients with diabetes.[27] A qualitative study has provided insight into the experiences of people diagnosed with various types of diabetes in the initial phase of lockdown in Denmark.[28] However, little is known about how people with T1DM handled daily life during the pandemic over a longer period of time that includes the reopening of society. Knowledge of the experiences and actions of people with T1DM during the COVID-19 pandemic may provide insight into strategies for handling the situation as well as possible needs for assistance from the healthcare system. This knowledge may in turn be valuable for other risk groups and in the event of similar situations in future. The aim of this study was to describe the experiences of adults with T1DM during the COVID-19 pandemic in Norway and what actions they took to cope with their situation.

## RESEARCH DESIGN AND METHODS
### Design and method description
This qualitative study used an inductive, descriptive design, applying the critical incident technique (CIT).[29] A 'critical incident' represents a major event of great importance to the person involved, and is an essential part of the CIT methodology. Critical incidents consist of positive or negative experiences in addition to actions carried out to handle the phenomenon in question.[30] Due to its flexibility and the focus on the patient perspective, CIT is considered an eligible method for use in healthcare research.[31] The method enables informants to describe significant incidents as well as their experiences and actions. Consequently, CIT is a suitable method for exploring the research question of this study.

### Participants and setting
The informants were recruited from the diabetes specialist outpatient clinics at three different hospitals in eastern Norway. The hospitals, two community hospitals and one university hospital, were situated in both rural and urban

areas. The total patient populations at the hospitals were 160 000, 240 000 and 560 000, respectively. People with T1DM will typically meet with a diabetes nurse or an endocrinologist in the diabetes outpatient clinics every three to 3–6 months.

A strategic selection was made to ensure a heterogeneous group of informants. Inclusion criteria were people with T1DM aged 18–65 years. Exclusion criteria were pregnancy, people with chronic pulmonary disorders, people with active cancer diseases and people diagnosed with a myocardial infarction or stroke during the previous 6 months. Characteristics such as gender, age, T1DM duration, metabolic control and having an occupation with increased exposure to possible infection were used in the selection of informants (table 1). Based on the criteria for strategic selection, a local designated diabetes nurse provided oral information about the study to potential participants. When interest was expressed, the nurse asked for oral consent to be contacted by the first author by telephone to provide further information about the study. The first author then made contact by telephone and gave additional information, and written information was sent to the potential participant by mail. After written informed consent was returned to the first author, an appointment for an interview was made. Three informants later cancelled the interview appointment due to a busy schedule (two) or illness (one). In total, 19 people with T1DM were included in the study.

### Data collection
Descriptions of experiences and actions were obtained through semistructured individual interviews. Due to the ongoing pandemic, informants could choose to conduct the interviews in person, by video call or by telephone. The video interviews were conducted on an encrypted web platform.

A semistructured interview was performed based on an interview guide. The interview guide was developed by all authors together, who have extensive clinical and methodological competence. A user representative from the local diabetes association reviewed the interview guide and gave advice on the content. After three interviews were conducted, the transcriptions were discussed within the research team. The interview guide consisted of three main topics and probing questions, as presented in table 2.

The interviews were conducted between July 2020 and December 2020. Norway was in complete lockdown from 12 March to mid-April 2020. After this, a cautious reopening of society was initiated. Nevertheless, the authorities' advice on social distancing and hygiene measures was maintained. In July 2020, when the first interviews were conducted, the community was relatively open. Later in the autumn, infection rates started to increase again, which led to a stronger focus on measures such as working from home, partially closed schools, the obligation to wear masks and so forth. There was no

**Table 1** Description of the participants of the study and characteristics of the interview situation

| Person | Age range | DM duration years | A1c mmol/mol | Diabetes complications | COVID-19* | Occupation exposed | Living alone | Interview form | Interview duration minutes | Site† |
|---|---|---|---|---|---|---|---|---|---|---|
| 1 | 60s | 46 | 61 | Yes | | | No | Meeting | 56 | 1 |
| 2 | 40s | 23 | 64 | Yes | | | No | Video | 25 | 1 |
| 3 | 30s | 23 | 51 | Yes | | | Yes | Meeting | 51 | 2 |
| 4 | 50s | 45 | 66 | Yes | | Kindergarten | Yes | Video | 39 | 2 |
| 5 | 20s | 17 | 60 | No | | | No | Video | 17 | 2 |
| 6 | 40s | 3 | 63 | No | (+) | | No | Video | 33 | 2 |
| 7 | 50s | 19 | 54 | No | | | No | Video | 59 | 1 |
| 8 | 20s | 25 | 50 | No | | | No | Video | 43 | 3 |
| 9 | 50s | 21 | 51 | No | + | | No | Video | 38 | 3 |
| 10 | 50s | 14 | 66 | No | + | | No | Video | 42 | 3 |
| 11 | 50s | 9 | 63 | No | | | No | Video | 22 | 3 |
| 12 | 40s | 29 | 53 | Yes | | School assistant | No | Video | 42 | 3 |
| 13 | 30s | 20 | 65 | No | | | Yes | Telephone | 57 | 3 |
| 14 | 20s | 10 | 46 | No | | Healthcare | Yes | Video | 27 | 3 |
| 15 | 50s | 53 | 57 | Yes | | | No | Video | 35 | 2 |
| 16 | 40s | 44 | 78 | No | | | No | Telephone | 45 | 2 |
| 17 | 40s | 37 | 53 | No | | Teacher | No | Video | 36 | 2 |
| 18 | 50s | 32 | 48 | Yes | | | No | Telephone | 93 | 2 |
| 19 | 50s | 26 | 53 | No | | | Yes | Video | 51 | 3 |
| Mean | 45 years | 26 years | 58 mmol/mol | | | | | | 43 min | |

*The symbol + denotes a confirmed case of COVID-19 infection, and the symbol (+) denotes suspected but unconfirmed infection.
†The three different sites of the study are represented by numbers 1, 2 and 3. The total patient populations at the three hospitals were 160 000 (1), 240 000 (2) and 560 000 (3). According to the annual report from the Norwegian Diabetes Register for Adults, 393 (1), 699 (2) and 1230 (3) patients with type 1 diabetes are treated and followed up at these outpatient clinics. According to the annual report from the Norwegian Diabetes Register for Adults, 393 (1), 699 (2) and 1230 (3) patients with type 1 diabetes are treated and followed up at these outpatient clinics.

**Table 2** Interview guide for the study, including probing questions and background information

| Topic | Q1 | Q2 | Q3 | Q4 | Q5 | Q6 | Probing questions | Background information |
|---|---|---|---|---|---|---|---|---|
| Topic 1: COVID-19 and daily life and coping with the situation | When you first heard about the COVID-19 virus and that it had spread to Norway, what were your thoughts? | Do you have any thoughts about whether this virus is dangerous for you, and if so, how have these thoughts affected you? | Have you taken any specific actions to take care of your diabetes treatment during this period? | Can you describe how your everyday life is or was affected by the COVID-19 pandemic? | Can you describe what it was like for you and your family when society started reopening? | Can you describe what it has been like for you to take care of the diabetes treatment during this time? | Where did this happen? What did you do in this situation/ can you explain more about what happened? What did you think in this situation? What feelings did you have in this situation? What do you think was the most demanding part of this situation? What does this mean for you in the future? | For how long have you had diabetes? Have you developed any diabetes complications? What was your last registered A1c? |
| Topic 2: sources of information about COVID-19 and diabetes | How did you and your family relate to news and media coverage of the pandemic? | Can you describe how you got information about the COVID-19 pandemic? | | | | | | |
| Topic 3: COVID-19 and help from the healthcare service | What expectations have you had about possible help/support from the health service to cope with the situation? | Do you feel that your expectations have been met? | If there was a new pandemic, what kind of health service would you wish to be established for people in risk groups? | | | | | |

current SARS-CoV2 vaccine available in Norway at the time of this study.

The interviews were all performed by the first author and recorded on a digital recorder. The first author transcribed 12 and a project assistant transcribed 7 of the audio recordings verbatim.

## Patient and public involvement
A user representative from the local diabetes association gave input in the preparation of the interview guide. Patients and the public were not involved in the development of the research question, design and conduct of this study. However, the results of the study will be disseminated on a local and/or national meeting in cooperation with the diabetes association.

## Data analysis
The study was analysed in line with CIT as described by Fridlund et al.[30] Data were regularly discussed in the research team until consensus regarding categorisation was achieved among the three authors. The first author read the interviews several times to obtain an overall picture of the content. Experiences and actions related to the purpose of the study were highlighted in the text and delimited. A total of 313 experiences and 185 actions were identified and categorised.

Delimitations from the text were then compared for similarities in an effort to group them into subcategories. When the subcategories were identified, a new process of exploration and comparison was performed to group the subcategories into categories. The purpose of a category is to describe the general character of the subcategories. Following the establishment of categories, a new effort was made to group categories into main areas. The purpose of constructing main areas was to describe the overall theme of the data. Descriptions of experiences and actions were analysed separately. This process led to the finding of 21 subcategories of experiences and 18 subcategories of actions. The subcategories were then structured into six categories of experiences and five categories of actions. Finally, two main areas of experiences and actions were identified.

## RESULTS
In total, 10 female and 9 male people with T1DM participated in the study, of which 2 had been diagnosed with COVID-19. All the informants were Norwegian born with the same cultural background (table 1).

## Experiences
Table 3 presents the experiences of people with T1DM during the COVID-19 pandemic in Norway, including direct quotations from the interviews. Two main areas emerged from the data analysis of experiences: 'increased psychosocial burden of T1DM during the COVID-19 pandemic' and 'changed conditions for T1DM treatment during the COVID-19 pandemic'.

## Increased psychosocial burden of T1DM during the COVID-19 pandemic
### Concerns related to COVID-19 and T1DM
Informants described uncertainty caused by belonging to a risk group. This included fear of getting sick from COVID-19 and confusion about the reasons why T1DM was associated with higher risk of a fatal outcome. General anxiety about the uncertainty of the overall situation was described as difficult.

### A need for knowledge about T1DM and COVID-19
The continuous media coverage caused anxiety and worries. The informants expressed a need for balanced information aimed at people in risk groups. Additionally, personalised information was desired and sought by contacting familiar healthcare providers such as the general practitioner or the diabetes nurse.

### Consequences for social relations as a result of COVID-19-related restrictions
This category describes the various difficulties the informants experienced while socialising with family, friends, colleagues and others. To avoid bringing the COVID-19 virus into the home, family members of people with T1DM were subjected to stricter restrictions than their peers. For instance, the children of one informant were not allowed to use public transport, and the son of a divorced woman moved temporarily to live with his father. Experiences of social limitations in the workplace due to the T1DM diagnosis were reported. For example, one of the informants was told that it was not desirable for her to return to the workplace after the termination of lockdown, as colleagues were afraid of infecting her.

## Changed conditions for T1DM treatment during the COVID-19 pandemic
### Altered routines due to COVID-19 affecting T1DM self-management
Restrictions implemented by the government to prevent the spread of COVID-19 created changes in daily routines, which in turn affected T1DM self-management. Informants expressed difficulties in maintaining a healthy lifestyle in terms of physical activity and healthy diet. To have a good time within the family, food such as cakes and ice cream were served more frequently. Since the gyms were closed, less familiar activities were initiated, which made it more difficult to predict correct insulin doses. Informants who contracted COVID-19 experienced large fluctuations in and high levels of blood glucose during the period of illness. Having access to continuous blood glucose monitoring (CGM) and insulin pumps was seen as an advantage in handling this situation. Reaching out to the diabetes nurse by telephone for advice on how to titrate insulin doses gave a sense of security.

### Changes in the work or study situation due to COVID-19 affecting T1DM self-management
Measures to prevent infection in the workplace led to new routines. This had an impact on meal breaks, time

**Table 3** Data analysis of experiences from quotations, grouped by subcategory, category and main area

| Quotation | Subcategory | Category | Main area |
|---|---|---|---|
| "Because it was so uncertain whether I was more at risk or, uhm yes, or how things were because no one knew that much. So, I thought it was scary". (3) | Uncertainty caused by belonging to a high-risk group | Concerns related to COVID-19 and T1DM | Increased psychosocial burden of T1DM during the COVID-19 pandemic |
| "But I was genuinely afraid of getting this, because I thought that if I get it, I'll die". (9) | Anxiety concerning infection | | |
| "The worst thing about the corona, when it came, it was all the uncertainty – we knew nothing". (18) | Concerns about the overall situation | | |
| "I paid attention at first, and then I thought this is not good, I will get depressed from seeing this, all this intimidating propaganda". (19) | Continuous media focus raised concerns | A need for knowledge about T1DM and COVID-19 | |
| "If I've been missing anything to some extent, it might getting insight into why diabetics are particularly vulnerable in a corona situation". (10) | Need for balanced information for people in risk groups | | |
| "I contacted the doctor's office. I also contacted the hospital to find out if I was at risk and what to do if called back to work". (4) | A desire for personalised information | | |
| "No, it's more the social discomfort you feel when people you meet sort of go for a hug and then you have to say 'You know what? I don't do that'". (14) | Challenges in social distancing | Consequences in social relations from COVID-19-related restrictions | |
| "And it was – especially for the children – it was probably the most demanding, that they had to avoid being with others for such a long time". (10) | Increased restrictions for family members of people with T1DM | | |
| "So, it was kind of that I should not return to work, so that they [her colleagues] would not get a bad conscience if I were to die". (4) | Restrictions related to being in a high-risk group due to T1DM | | |
| "I felt, when we locked down, that … that the start was completely awful because it, uhm, it completely changed my routines". (3) | Difficulties creating routines in everyday life at home | Altered routines due to COVID-19 affected T1DM self-management | Changed conditions for T1DM treatment during the COVID-19 pandemic |
| "Uhm yes, but it makes one stay indoors more and such, and then it's easier to make a little extra-good food so, … "(2) | Difficulties maintaining a healthy lifestyle | | |
| "Surprisingly, the treatment has been better than ever before. But it has to do with the fact that I got a new type of insulin pump. And it has really made everyday life a lot easier". (8) | The benefit of glucose sensor and insulin pump in diabetes treatment | | |
| "I had an incredibly high fever and had an insanely high blood glucose". (9) | Challenging fluctuations in blood glucose during COVID-19 disease | | |
| "It [the blood glucose] has been affected because I, well it's much busier at work [teacher] because we wash desks and surfaces all the time". (17) | New routines at work to prevent the spread of COVID-19 | Changes in the work or study situation due to COVID-19 affecting T1DM self-management | |
| "But clearly, the lack of breaks and not being able to eat and such, I must say that I have sometimes felt that it has been tough at work". (17) | Increased workload due to COVID-19 caused stress | | |
| "All the time I feel that I have kind of a 'sting' in the back of my mind, whether I might be laid off again". (18) | Financial uncertainty caused stress | | |
| "It's even easier, when I work from home, to maintain good blood glucose control, compared with running around taking those measurements at work all the time". (10) | More predictable everyday life gave more time to focus on T1DM self-management | | |
| "The one [health check] before the summer was just completely eliminated, so there was a year in between check-ups and I don't think that is good". (13) | Lack of or delayed follow-up from the specialist health service | Reorganisation of the health service due to COVID-19 affecting T1DM treatment | |

Continued

**Table 3** Continued

| Quotation | Subcategory | Category | Main area |
|---|---|---|---|
| "I must admit that in the beginning I thought 'This is a state of emergency, now I have to stock up on insulin'". (15) | Concerns regarding access to medicines | | |
| "And if I should end up with someone who does not necessarily have full control over what blood glucose is all about in such an intensive ward, then yes". (7) | Concerns about intensive care unit healthcare professionals' knowledge of specialised diabetes treatment in case of severe illness and need for help | | |
| "In the end I called the diabetes nurse and asked: 'What am I going to do? I can't get my blood glucose down'". (9) | Availability of healthcare professionals by telephone | | |

T1DM, type 1 diabetes.

spent being physically active and opportunities for social contact at work. Some, such as a teacher and a mail delivery worker, experienced increased work pressure and new work tasks, whereas others experienced a more predictable daily life with less stress and more time to take care of T1DM self-management. Being laid off from work because of lockdown gave rise to financial insecurity. This led to stress, which in turn affected blood glucose levels.

### Reorganisation of the health service due to COVID-19 affecting T1DM-treatment

Informants experienced delayed follow-up from the specialist outpatient clinic. This was due to the reorganisation of hospital resources to prepare for a large number of patients with COVID-19. The authorities introduced a temporary rationing of insulin to prevent hoarding and ensure access. This led to concerns about the availability of medications and created worries about the inconvenience of having to go to the pharmacy more frequently. Informants' concerns about possible lack of competence in the intensive care unit regarding specialised T1DM treatment made the risk of becoming severely ill from COVID-19 even more frightening. These concerns related to previous experiences from hospitalisations. Having experienced easy access to healthcare professionals before the pandemic made them feel confident that they could reconnect for information and advice when needed.

### Actions

Table 4 presents the actions of people with T1DM during the COVID-19 pandemic in Norway, including direct quotations from the interviews. The two main areas of actions that emerged from the data analysis were 'actions to handle psychosocial strain related to T1DM and COVID-19' and 'actions to handle changed conditions for T1DM treatment during the COVID-19 pandemic'. The analysis of actions is described in table 4.

### Actions to handle psychosocial strain related to T1DM and COVID-19

#### Practical strategies to avoid being infected with COVID-19

Informants described increased awareness of complying with general advice on infection control. Measures such as social distancing and frequent hand washing became a daily routine. Some started using a mask early on. Avoiding crowded places and receiving help from others for grocery shopping and providing necessary medications were strategies adopted to prevent infection. Limiting the number of social contacts for the family as a whole was considered necessary to avoid infection.

#### Measures to maintain social contact

Informants described finding support and joy within the family. Choosing a few people to meet regularly gave them the opportunity to maintain social contact outside the home. Children of parents with T1DM were allowed to choose one or two friends to play with. Both new and familiar digital platforms served as alternative arenas for social interaction.

#### Mental strategies to deal with the situation

Informants sought personalised information about their risk status. Positive thinking was one of the main strategies to cope with the overall situation. Furthermore, informants actively avoided media coverage of COVID-19.

### Actions to handle changed conditions for T1DM treatment during the COVID-19 pandemic

#### Creating a safe structure in daily life

Predictable routines were created to facilitate T1DM self-management. New job routines were implemented to protect the informants from being infected. Working from home was challenging, but establishing home-office routines was experienced as a positive factor. Lack of routine made handling work and T1DM self-management difficult. Meaningful activities at home were found helpful to avoid pondering about the situation.

**Table 4** Data analysis of actions from quotations, to subcategories, via categories up to the main areas

| Quotation | Subcategory | Category | Main area |
|---|---|---|---|
| "I was very, very conscious about keeping my distance to others, and as I said, I wore a mask as well". (18) | Increased awareness of complying with infection control advice | Practical strategies to avoid being infected with COVID-19 | Actions to handle psychosocial strain related to T1DM and COVID-19 |
| "Every Saturday he [her brother] goes to the store and buys groceries for me". (15) | Getting help from others to obtain food and medicine | | |
| "We can't let you [the children] take the bus unnecessarily, you should not go to a party where there are 50 people, that is not what we should do now. So yes, they must consider me". (16) | Limiting family members' number of social contacts | | |
| "I have a good friend who has bone marrow cancer. And we both agreed to it because she is very restrictive about who she socializes with too, so the two of us had a lot of contact". (4) | Choosing a few people to meet and socialise with | Measures to maintain social contact with others | |
| "And in this hobby community they are good at organizing such online meetings, so we sit at our PCs together instead of alone". (12) | Creating alternative arenas for socialisation | | |
| "I'm so fortunate that I have a husband that I live with. I'm married, you know, so I think that the loneliness that many feel. I'm much better off". (15) | Finding joy and support within the family | | |
| "And then I thought well, it's not certain that it's really bad that I get it (COVID-19) because I'm pretty healthy, I have a good Hba1c, my immune system is kind of good". (5) | Positive cognitive reflections to deal with the situation | Mental strategies to deal with the situation | |
| "I kind of tried to avoid maybe the newspapers as much as possible, because I feel that … well, it affects me in a negative sense". (3) | Selective attitude towards media information | | |
| "I asked my doctor what should I do? Should I just continue working from home and answer technical questions, or should I travel and work as normal?" (19) | Actively searching for personalised information | | |
| "I quickly decided that I must live as normally as possible. Eat as normal. Be physically active outside. Keep in touch with others in a safe way". (4) | Creating predictable everyday routines for T1DM self-management | Creating a safe structure in daily life | Actions to handle changed conditions for T1DM treatment during the COVID-19 pandemic |
| "So then I eat in the car- instead of going in there to eat with the others [on a construction site]". (2) | Starting new routines at work for infection control | | |
| "I've been pretending that it's a normal day. I get up at regular times, shower, put on make-up, etc". (15) | Creating appropriate home-office routines | | |
| "And I have enough things to do so that there will not be much effort spent on thinking too much, to put it that way". (11) | Consciously finding meaningful pursuits to avoid brooding | | |
| "And also I think we are very fortunate to have the diabetes system we have here in Norway; that I was allowed to get Freestyle Libre and such". (10) | Using medical technical equipment actively in treating diabetes | Measures to ensure proper T1DM self-management during the COVID-19 pandemic | |
| "I could not go to the gym, so I had to use nature instead". (4) | Alternative physical activities to maintain a healthy lifestyle during the COVID-19 pandemic | | |
| "And tried to inject, took higher and higher insulin doses". (9) | Titrate insulin doses in relation to blood glucose fluctuations during COVID-19 disease | | |

Continued

**Table 4** Continued

| Quotation | Subcategory | Category | Main area |
|---|---|---|---|
| "Then I was isolated in the bedroom by my wife. I was not allowed to go out". (9) | Isolating due to COVID-19 illness | | |
| "And about medicine and stuff. I have always had a stock of insulin". (4) | Keeping a stockpile of insulin available for emergencies | | |

T1DM, type 1 diabetes.

### Measures to ensure proper T1DM self-management during the COVID-19 pandemic

Medical devices such as CGM and insulin pumps were perceived as useful in the treatment of T1DM and contributed to good metabolic control. Furthermore, informants found alternative physical activities to maintain a healthy lifestyle when the fitness centres were closed. Keeping a storage of insulin led to a sense of security. For those who contracted COVID-19, high blood glucose levels caused by the infection were met with efforts to titrate insulin doses, and informants called the diabetes nurse for support and advice on this matter.

### CONCLUSION

This study aimed to describe the experiences and actions of people with T1DM during the COVID-19 pandemic in Norway between July and December 2020, when no current SARS-CoV2 vaccine was available in Norway. Results indicate that the psychosocial burden of living with T1DM increased during the pandemic. Uncertainty caused by belonging to a risk group, fear of severe illness and strict adherence to social distancing for the family as a whole contributed to the burden of T1DM. Additionally, we found that infection control measures and reorganisation of the healthcare system represented challenges in the form of changed conditions for T1DM treatment. Applying manageable, understandable and meaningful actions to deal with the psychosocial strain and the disrupted daily life led to a general sense of coping despite difficulties.

Alteration of daily routines brought challenges such as an increased amount of unhealthy food and less exercise. These findings are in line with previous quantitative research.[32 33] In a Saudi cross-sectional study of people with T1DM using an insulin pump, difficulties in maintaining a healthy diet and staying physically active during lockdown were seen. However, adherence to the insulin pump therapy contributed to stabilising blood glucose.[32] Participants in our study described that general stress and anxiety affected T1DM self-management in terms of fluctuations in blood glucose similar to those seen in studies from Italy.[22 34] Nevertheless, our participants also described a more predictable lifestyle in working from home that led to improved control of blood glucose levels. Similar findings of improved metabolic control have been reported from studies of children with T1DM on insulin pump therapy.[35–38] Furthermore, multiple studies that reviewed metabolic control via CGM data registered during the pandemic found improvements in both time in range and blood glucose levels,[23–25] although one study showed a tendency of increased fluctuations in physical activity and perceived stress.[34]

The finding of an increased psychosocial burden of T1DM during the COVID-19 pandemic is consistent with existing evidence from quantitative studies.[26 27] Our study provides an in-depth understanding of these psychosocial difficulties, and is consistent with a qualitative study conducted in Denmark.[28] While the Danish study focused on the experiences of people with multiple types of diabetes during lockdown in Denmark (April 2020), the results from our study focused on people with T1DM at a later stage of the pandemic (July 2020 to December 2020). Therefore, we covered experiences related to the reopening of society as well as to new periods of lockdown. In comparison with the Danish study, our findings may indicate that the psychosocial burden of T1DM during the pandemic increased over time. This interpretation is supported by another qualitative study of people living with different long-term physical health conditions during the COVID-19 pandemic. In this study they found that the COVID-19 pandemic affected several aspects of mental health and well-being.[16] Furthermore, in a review of the effects of quarantine, impacts on psychosocial health were seen, and the length of the period of isolation affected the outcomes of these negative effects.[12] Even though our participants were not quarantined, they were to some extent self-isolating for extended periods, and thus the psychosocial burden might have increased accordingly.

'Diabetes distress' refers to a psychological condition associated with the burden of living with T1DM.[39] Diabetes distress might add to general stress from the severity of the pandemic situation and further increase the psychosocial burden of T1DM. In a previous study, diabetes distress was identified as one of several factors associated with poorer psychosocial health in patients with diabetes during the COVID-19 pandemic.[26]

Freeston *et al*[40] have proposed a model of uncertainty distress in the context of the COVID-19 pandemic. According to this model, uncertainty can be defined as 'the subjective negative emotions experienced in response to the as yet unknown aspects of a given situation'.[40] The degree of uncertainty distress depends on the following factors: disruption of everyday life, perceived

actual threat and perceived uncertainty. In addition, individual tolerance of uncertainty influences uncertainty distress. Individual tolerance of uncertainty can be modified by uncertainty-reducing behaviours. Such behaviours may include actions such as avoidance, preparation for possible negative consequences, postponement and the search for reassurance. The actions taken by the participants in our study can be seen in the light of this model of uncertainty distress in the context of the COVID-19 pandemic. Our participants actively avoided media, they prepared for possible negative consequences by keeping a stock of insulin and by strictly complying with advices on infection control measures. Furthermore, social gatherings were postponed, and the participants searched for tailored and personalised information for reassurance.

In this study, we have described how various actions were taken by the participants to adapt, and positive thinking was one of the main strategies to cope with the overall situation. From the perspective of the theory of salutogenesis, stressors in life will not necessarily contribute to poor outcomes of health if met properly. The essential factor is that the individual finds understandable, manageable and meaningful actions to adapt. Through meaningful actions, people may achieve a sense of coherence, and thus of coping despite challenging environments.[41–43]

Participants in our study emphasised the importance of access to healthcare professionals by telephone for personalised guidance. A recently published but not yet peer-reviewed study from Norway shows a surprising and significant increase in non-COVID-related deaths in patients with diabetes from March 2020 to May 2020.[44] The causal factors behind these findings are unknown, and it is not possible to draw conclusions. Nevertheless, Raknes *et al*[44] propose inappropriate avoidance of healthcare professionals and thus a lack of follow-up as a possible explanation. Based on this understanding, the importance of having healthcare professionals such as diabetes nurses available by telephone for medical advice and social support is further underlined.

### Study limitations

We have strived to achieve trustworthiness throughout the study in terms of credibility, dependability, confirmability and transferability as described by Polit and Beck.[45] Three researchers with good knowledge of both CIT and diabetes have been involved throughout the study. The first is a diabetes specialist nurse, she has been involved in all stages of this research project. The two others are professors in nursing with extensive methodological knowledge of CIT.

The first author has clinical experience, which includes answering calls from diabetes patients anxious about COVID-19. Her experience represents a factor that might threaten reflexivity. On the other hand, substantial knowledge from the field can be a valuable source of relevant and specific research.[46] The first author's clinical experience can be seen as an advantage in the process of formulating relevant questions for the interview guide.

Awareness of the potential influence of previous experiences has been taken into account in the interpretation of the data. Description of the data has sought to be clear and has been discussed in depth within the research team to strengthen the credibility of the study. The interviews were all performed by the first author, which increased the likelihood that the interview guide was used in the same way in all the interviews. Out of 19 interviews, 14 were conducted by video call and three were conducted by telephone, which may have contributed to a loss of some nonverbal data.[47] On the other hand, participants had the comfort of familiar environments and did not have to worry about possible infection. We have provided a detailed description of the participants as important contextual factors regarding the dependability of the study (table 1). In sampling, we aimed to achieve variation in gender, age, metabolic control, T1DM duration and occupation. However, we were unable to include participants with a different cultural background, which is a limitation. A possible paradox in the study may be that subjects who suffer most from lockdown, could be the ones who want to talk and seek comfort from healthcare providers. To strengthen dependability and facilitate confirmability, the systematic procedure we followed in the data analysis is clearly presented with quotations categorised and finally grouped into main areas (tables 3 and 4). A diversity of experiences and actions were obtained from the interviews, and this is seen as a strength when using the CIT approach.[29]

According to Malterud, the field and the context in which the data material is acquired will always determine the scope of the knowledge in time and space.[48] From an international perspective, Norway has had relatively few deaths from COVID-19 compared with other European countries.[49] As of the 28 April 2021, there are 753 registered deaths out of 111 685 confirmed COVID-19 cases in a population of 5 391 369 inhabitants.[50] Participants included in our study live in the eastern part of Norway, close to the capital. The capital and surrounding areas are the part of Norway most affected by the COVID-19 pandemic in terms of lockdown restrictions. From this perspective, it would be fair to say that the participants in this study may have had experiences similar to people in countries with higher infection rates and deaths from COVID-19. A detailed description of the study context, the participants, method of data collection and the data analysis, including direct quotations from the interviews, are presented (tables 1–4) to give the reader the opportunity to determine the transferability of the findings.

### Conclusion and implications for clinical practice

In conclusion, participants experienced psychosocial strain and an increased burden of T1DM. Furthermore, they described uncertainty-reducing behaviours and actions, which provided a general sense of coping with handling a disrupted daily life during the COVID-19 pandemic despite difficulties.

Healthcare professionals should be aware that many people with T1DM might have lived under increased stress during the COVID-19 pandemic. Availability of healthcare professionals for personalised advice by telephone was emphasised as important to alleviate uncertainty distress, reduce diabetes distress and support adequate T1DM self-management. Access to telephone and video consultations should be implemented to ensure follow-up of people with T1DM in similar situations in future.

Further research is needed to investigate potential psychosocial impacts of the COVID-19 pandemic on people with T1DM from a family perspective. Exploring possible associations between diabetes distress, uncertainty distress and psychosocial burden was beyond the scope of this study. However, this would be an interesting topic for future research.

**Acknowledgements** The authors thank the patient representative May Kvernstuen for valuable input in preparation to the interview guide. This study was partly funded by Viken County Diabetes Association.

**Contributors** KP, AB and MMI designed the study. MMI facilitated data collection. KP collected the data. KP, AB and MMI contributed to data analysis. KP, AB and MMI contributed to drafting the manuscript and read and approved the final manuscript. MMI is the guarantor for this study.

**Funding** This study was funded by Diabetesforbundet.

**Competing interests** KP has received for a contribution of NOK 10 000 from the regional Diabetes Association for the project.

**Patient and public involvement** Patients and/or the public were involved in the design, or conduct, or reporting, or dissemination plans of this research. Refer to the Research design and methods section for further details.

**Patient consent for publication** Not applicable.

**Ethics approval** This study involves human participants and was approved by the Regional Committe for Medical and Health Research Ethics in South East Norway (REC-154546), the Norwegian Centre for Research Data (NSD) (NSD-39284), the local data protection officers at the three study sites and Western Norway University of Applied Science before study start. Participants gave informed consent to participate in the study before taking part.

**Provenance and peer review** Not commissioned; externally peer reviewed.

**Data availability statement** All data relevant to the study are included in the article or uploaded as supplementary information. N/A.

**ORCID iD**
Marjolein Memelink Iversen http://orcid.org/0000-0001-9954-171X

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
