## [Reviewer comments · BMJ Open]

ARTICLE DETAILS

TITLE (PROVISIONAL)	Experiences and actions related to living with type 1 diabetes during the COVID-19 pandemic in Norway - A qualitative study conducted during July - December 2020
AUTHORS	Pleym, Karin; Iversen, Marjolein; Broström, Anders

VERSION 1 – REVIEW

REVIEWER	Harbitz, Martin University of Tromsø Department of Community Medicine
REVIEW RETURNED	30-Aug-2021

GENERAL COMMENTS	My first impression is an structured and well written paper on an important topic. It is a qualitative interview study on a strategic selection of diabetes type 1 (T1DM) patients and their experiences during the covid 19 pandemic. The authors have used the Critical Incident Technique to gather data about significant events and patients' experiences and actions. Structure Is the article clearly laid out? Yes Title: Does it clearly describe the article? Yes By section: Abstract: Does it reflect the content of the article? Yes Introduction Well written and structured. Design and methods Data analysis: Reading the paper several times it looks like the first author has done all the interviews and all the analysis work herself. The analysis process is well described, but there should have been more researchers present during both interviews and analysis. The researchers may have done this, but it is not easy to read as the text stands now. If there was only one researcher doing the analysis this is a weakness and deserves more explanation. Results: The results are presented clearly and well organized. Well done. Discussion: The discussion section is interesting and well written.
--

	Study limitations: The abovementioned weakness is addressed here. Though it could be described more detailed and informative. For example: "Several researchers with good knowledge of both CIT and diabetes have been involved throughout the study". Who were these researchers? And what exactly did they contribute with? There are three more authors in this paper, considering reflexivity issues, their contribution need to be described
--	---

REVIEWER	D'Onofrio, Luca University of Rome La Sapienza
REVIEW RETURNED	28-Oct-2021

GENERAL COMMENTS	The study conducted by Karin P and coll aimed to "describe the experiences of adults with T1DM during the COVID-19 pandemic in Norway and what actions they took to cope with their situation" applying the critical incident technique (CIT). The aim was interesting and cover a debated field of clinical research. Further, the authors used an interesting qualitative method to collect the data presented. The results are in line with the aim of the study. However, minor issues arose reading the manuscript, and for this reason, the manuscript needs some revision. MINOR revision:  1) The authors stated that "The interviews were performed between July 2020 and December 2020." It is very important for the readers of this manuscript to understand the condition at the moment of the study conduction. The authors should explain what type of lock-down was in place at the time of the study and whether there had been more stringent measures previously. 2) The study period should be specified also in the title. 3) In the discussion, the authors should highlight that this evidence is gathered before the Sars-Cov2 vaccine use. 4) The authors have not considered a possible paradox in the study participation. Subjects who suffer most from the lock-down, usually could be the ones who most want to talk and seek comfort from healthcare providers. The authors should address this possible bias/limitation in the discussion. 5) Comment also the following studies that showed that patients with T1D improved their glycaemic control during pandemic giving more insight (doi.org/10.2337/dc20-0954, doi.org/10.2337/dc20-1127). 6) patients with T2D that mentally suffer most during the pandemic, showed a worse metabolic control (refer to: doi.org/10.1111/dom.14380), comment if this consideration is valid also for patients with T1D. Consider to check HbA1c for patients included in the analysis before and after the lock-down period.
--

VERSION 1 – AUTHOR RESPONSE

REVIEWER 1 (Mr. Martin Harbitz, University of Tromso Department of Community Medicine)

Comments to the Author:

My first impression is a structured and well written paper on an important topic.

It is a qualitative interview study on a strategic selection of diabetes type 1 (T1DM) patients and their experiences during the covid 19 pandemic. The authors have used the Critical Incident Technique to gather data about significant events and patients' experiences and actions.

Response:

Thank you for your comment, we are grateful for the acknowledgement of our study.

1. Reading the paper several times it looks like the first author has done all the interviews and all the analysis work herself. The analysis process is well described, but there should have been more researchers present during both interviews and analysis. The researchers may have done this, but it is not easy to read as the text stands now. If there was only one researcher doing the analysis this is a weakness and deserves more explanation.

Response:

Thank you for this important comment. It is correct that all the interviews were performed by the first author. However, the interview guide was developed by all the three authors together, who has extensive clinical and methodological competence. The outcome of the initial interviews was carefully evaluated and discussed, both regarding the design of the interview guide and the approach of the interviewer before deciding to use the interviews. All three authors have been active in the analysis of the data. Data was regularly discussed in the research team until consensus was achieved among the three authors. We have now clarified this in the manuscript (Page 6/ Para 4; Page 7/ Para 3; Page 14/ Para 4).

2. The discussion section is interesting and well written.

Response:

Thank you for your comment.

3. Study limitations: The abovementioned weakness is addressed here. Though it could be described more detailed and informative. For example: "Several researchers with good knowledge of both CIT and diabetes have been involved throughout the study" (Page 15 / Para 4 / Line xx). Who were these researchers? And what exactly did they contribute with? There are three more authors in this paper, considering reflexivity issues, their contribution need to be described

Response:

Thank you for pointing this out. In total we are three researchers involved in this study, a diabetes specialist nurse, she has been involved in all stages of this research project. The two others are professors in nursing with extensive methodological knowledge of CIT, who also participated in the design of the study. The interview guide was developed by all authors together, who have extensive clinical and methodological competence. Methodological aspects of the analysis, the transcription and initial review of the interviews, the identification of experiences and actions, as well as the categorization were regularly discussed in the research team until consensus was achieved among the three authors. The researchers clinical and methodological experience were used in this process. We have added some more information on this (a bit shorter due to word limitations in the manuscript) to clarify the role of the researchers and addressed their competence, their role and reflexivity issues in the manuscript (Page 6/ Para 4; Page 7/ Para 3; Page 14/ Para 4; Page 15/ Para 2).

REVIEWER 2 (Dr. Luca D'Onofrio, University of Rome La Sapienza)

Comments to the Author:

The study conducted by Karin P and coll aimed to "describe the experiences of adults with T1DM during the COVID-19 pandemic in Norway and what actions they took to cope with their situation" applying the critical incident technique (CIT). The aim was interesting and cover a debated field of clinical research. Further, the authors used an interesting qualitative method to collect the data presented. The results are in line with the aim of the study. However, minor issues arose reading the manuscript, and for this reason, the manuscript needs some revision.

Response:

Thank you for your comment

MINOR revision:

1. The authors stated that “The interviews were performed between July 2020 and December 2020.” It is very important for the readers of this manuscript to understand the condition at the moment of the study conduction. The authors should explain what type of lock-down was in place at the time of the study and whether there had been more stringent measures previously.

Response:

Thank you for this comment. We have revised the manuscript and included information regarding the lock-down situation for the time-period when interviews were conducted (Page 7/ Para 1; Page 12/ Para 2; Page 15/ Para 3; Page 16/ Para 1).

2. The study period should be specified also in the title.

Response:

We have updated this issue in the revised manuscript.

3. In the discussion, the authors should highlight that this evidence is gathered before the Sars-Cov2 vaccine use.

Response:

We have updated this issue in both the method and discussion of the revised manuscript (Page 7/ Para 1; Page 12/ Para 2/ Line 2-3).

4. The authors have not considered a possible paradox in the study participation. Subjects who suffer most from the lock-down, usually could be the ones who most want to talk and seek comfort from healthcare providers. The authors should address this possible bias/limitation in the discussion.

Response:

Thank you for your important comment. We have now specified this possible bias in the limitation section (Page 15/ Para 2)

5. Comment also the following studies that showed that patients with T1D improved their glycaemic control during pandemic giving more insight (doi.org/10.2337/dc20-0954, doi.org/10.2337/dc20-1127).

Response:

Thank you for this comment, we have now included these two studies.

6. Patients with T2D that mentally suffer most during the pandemic, showed a worse metabolic control (refer to: doi.org/10.1111/dom.14380), comment if this consideration is valid also for patients with T1D. Consider to check HbA1c for patients included in the analysis before and after the lock-down period.

Response:

Thanks for this important comment. We have considered this aspect and decided not to include HBA1c. The study uses a qualitative design and describe experiences of adults with type 1 diabetes and what actions they took to cope with the situation during the COVID-19 pandemic in Norway. The sample is described by clinical and demographical variables often used in studies with this design. All these variables were collected before the interviews. In a qualitative study all data, in this case

experiences and actions described by the participants are of the same importance. Linking a pre or post HBA1c value from one patient to a specific finding is not important. The information given is perceived as sufficient to get a good understanding of the sample. Therefore, we consider adding a post value for HBA1c would be of marginal importance.

Editorial requests:

1. Please revise your title so that it includes your study design, rather than the type of method/ analysis used. This is the preferred format for the journal. We suggest: "Experiences and actions related to living with type 1 diabetes during the COVID-19 pandemic in Norway: A qualitative study."

Response:

Thank you for your comment. We have now changed the title. The new title reads "Experiences and actions related to living with type 1 diabetes during the COVID-19 pandemic in Norway - A qualitative study conducted during July - December 2020".

2. Please revise the abstract so that it is following the structured abstract recommended in BMJ Open's Instructions for Authors for research articles. See: <https://bmjopen.bmj.com/pages/authors/#research>

Response:

We adjusted our abstract accordingly. Please, let us know if we do not adhere to the recommendations of BMJ Open.

3. Please revise the strengths and limitations section after the abstract. It should contain up to five short bullet points, no longer than one sentence each, that relate specifically to the methods of the study reported (see: <http://bmjopen.bmj.com/site/about/guidelines.xhtml#articletypes>) . It should be clear why each point is a methodological strength or limitation. Currently, it's not clear if there are any limitations in this section.

Response:

Thank you for this response. We have now adjusted the strength and limitation section after the abstract. Please, let us know if we do not adhere to the recommendations of BMJ Open.

4. Page 5: "The study protocol was evaluated by the Regional Committee for Medical and Health." Can you please make it clear that these committees *approved* (not just evaluated) your study? Please also check the paper for typographical errors e.g. 'Committe' should be 'Committee'

Response:

The evaluation of the study protocol followed ordinary recommendations. Firstly, it was evaluated by the Regional Committee for Medical and Health Research Ethics in South East Norway (REC-154546). The evaluators concluded that only approval was needed from the local data protection officers at the three study sites and Western Norway University of Applied Science before study start. All four data protection officers approved the study before study start. We have added the word

“accepted” in the manuscript, but not described the above details in the text to avoid confusion. (Page 5/ Para 3).

VERSION 2 – REVIEW

REVIEWER	D'Onofrio, Luca University of Rome La Sapienza
REVIEW RETURNED	24-Dec-2021
GENERAL COMMENTS	All the requested revisions were addressed